# Characterization of clopyralid resistance in lawn burweed (*Soliva sessilis*)

**Hossein Ghanizadeh**[1]*, **Fengshuo Li**[2], **Lulu He**[1], **Kerry C. Harrington**[1]

**1** School of Agriculture and Environment, Massey University, Palmerston North, New Zealand, **2** College of Horticulture and Landscape Architecture, Northeast Agricultural University, Harbin, China

\* H.Ghanizadeh@massey.ac.nz

## Abstract

*Soliva sessilis* is a troublesome annual weed species in New Zealand turfgrass. This weed has been controlled selectively in New Zealand turfgrass for many years using pyridine herbicides such as clopyralid. However, in some golf courses, the continuous application of pyridine herbicides has resulted in the selection of *S. sessilis* populations that are resistant to these herbicides. This study focuses on a clopyralid-resistant population of *S. sessilis* collected from a golf course with a long history of clopyralid applications. The resistant phenotype of *S. sessilis* was highly resistant to clopyralid (over 225-fold). It was also cross-resistant to dicamba, MCPA and picloram but not mecoprop. The level of resistance to dicamba was high (7-14-fold) but much lower (2-3-fold) for both MCPA and picloram. The phenotype was morphologically distinct from its susceptible counterpart. Individuals of the clopyralid-resistant phenotype had fewer lobes on their leaves and were slightly larger compared to the susceptible phenotype. Resistant individuals also had a larger leaf area and greater root dry weight than the susceptible plants. An evaluation of internal transcribed spacer (ITS) regions confirmed that clopyralid-resistant phenotypes are conspecific with *S. sessilis*. In summary, the cross-resistance to several auxinic herbicides in this *S. sessilis* phenotype greatly reduces chemical options for controlling it; thus, other integrated management practices may be needed such as using turfgrass competition to reduce weed germination. However, the morphological differences between resistant and susceptible plants make it easy to see, which will help with its management.

## Introduction

Weeds are unwanted plant species that can be troublesome in agricultural and non-agricultural situations [1]. Since the first herbicide was commercialized, chemical weed control has been the preferred method for managing weed populations [2]. The popularity of herbicides for weed control has not been without consequences and as predicted in the early days of the commercialization of herbicides, the occurrence of herbicide resistance in weed populations was inevitable [3]. To date there are over 500 unique cases of herbicide-resistant weed species globally [4]. In New Zealand, currently there are 25 confirmed cases of herbicide-resistant weeds [5], two of which were reported in turfgrass [6, 7].

*Soliva sessilis*, a member of the Asteraceae family, is a low growing winter annual weed species [8]. *S. sessilis* is originally from South America [9], and some of its common names include

**Funding:** Financial support for the authors was provided by the Endeavour fund (C10X1806, Managing Herbicide Resistance) from the New

Zealand Ministry for Business Innovation and
Employment. The funder did not play any role in
the study design, data collection and analysis,
decision to publish, or preparation of the
manuscript.

**Competing interests:** The authors have declared
that no competing interests exist.

lawn burweed, field burrweed, lawnweed, bindii, bindy-eye, carpet burweed and Onehunga
weed. In New Zealand, *S. sessilis* is primarily a troublesome weed species in turfgrass [5]. The
germination of *S. sessilis* seeds occurs in late summer or early autumn when the soil becomes
moist, and turfgrass has not recovered from summer dieback. Once the *S. sessilis* plants are
established, they grow throughout winter [9]. There is little known about the reproductive
biology of *S. sessilis*, but it has been noted that *S. sessilis* produces seeds in spring and, in sum-
mer when the mature plants of *S. sessilis* die back, the seeds are shed on the soil surface [9].
Each seed has a sharp spine that can penetrate the skin of bare feet, thus making this species a
nuisance in turfgrass areas, especially near playgrounds and in home lawns [10]. Also, *S. sessilis*
can be a vigorous competitor in short turfgrass, and once the plants die back, they leave
patches of bare soil that can be used as a niche for other weed species establishment [11].

Weed management practices for turfgrass weed species such as *S. sessilis* involve mechani-
cal, cultural and chemical options [12]. Hand pulling of established plants was found effective
since *S. sessilis* plants are shallow-rooted and can easily be pulled out [13]. However, this
approach is labour-intensive and is not cost-effective in large areas of turfgrass. Mowing does
not affect *S. sessilis* due to the prostrate growth habit of *S. sessilis*. Flaming as a cultural practice
was found effective in reducing *S. sessilis* populations [13]; however, the best results can only
be achieved by using high intensity fires [14], which are damaging to turfgrass. Keeping turf-
grass dense during autumn when the seeds normally germinate will prevent a new cohort
from establishing, but this weed causes problems in turfgrass that has died back due to summer
dryness [10]. Compared to other weed control practices, chemical options are more desirable
as they provide more efficient and selective control of *S. sessilis*, hence they have been the most
common practice for *S. sessilis* control in turfgrass in New Zealand [5].

Selective control of *S. sessilis* in New Zealand occasionally makes use of contact herbicides
such as bentazone or ioxynil for seedlings, but generally involves synthetic auxin herbicides
such as clopyralid, triclopyr, picloram, dicamba and MCPA for older plants [15]. Synthetic
auxin herbicides mimic the natural auxin indole-3-acetic acid (IAA), hence they can bind to
the same target receptor as IAA [16–18]. Clopyralid has been widely used to control *S. sessilis*
in turfgrass in New Zealand because of its effectiveness and selectivity in fine turfgrass [5, 6].
However, in the early 2000s, a population of *S. sessilis* was found resistant to clopyralid in a
New Zealand golf course which had a long history of pyridine herbicide applications, especially
clopyralid and triclopyr [6]. Worldwide, there are currently over 40 weed species confirmed as
having evolved resistance to synthetic auxin herbicides [4]. We understand this is the only
reported case in the world of clopyralid-resistant *S. sessilis*.

The initial report of this case showed only that there were significant differences between
this phenotype and susceptible *S. sessilis* to recommended rates of clopyralid, triclopyr, and
mixtures of picloram with triclopyr and with 2,4-D [6]. In work reported here, we evaluate the
magnitude of resistance to clopyralid in this phenotype, and also investigate the magnitude of
resistance within some of the other synthetic auxin herbicides used in turfgrass. Also, we pro-
vide further details about the differences in growth traits between clopyralid-resistant and clo-
pyralid-susceptible phenotypes of *S. sessilis*, and check whether this phenotype is still
genetically the same species given the differences in morphology.

## Materials and methods

### Plant material

The response of a clopyralid-resistant population of *S. sessilis* was compared to a susceptible
one. The clopyralid-resistant population (OR) was the population mentioned above which also
showed some resistance to triclopyr and picloram, and was originally from a golf course at

Helensville (36˚38'27.9"S 174˚30'43.0"E) near Auckland [6]. Seeds were collected from survivors of plants that had been treated with clopyralid and were kept at 5˚C. To multiply these seeds, 20 plants were grown from seeds that had been collected after the preliminary experiment. For this, five seeds were placed on the surface of potting mix (20% Pacific Pumice (7 mm), 30% fibre, 50% bark) containing slow release fertilizer (Woodace, PA, USA) within polythene bags, and the seeds were covered with a 1cm layer of potting mix. The pots were placed in a heated glasshouse at average daily max/min temperatures of 22.2/20.5˚C, and an average relative humidity (RH) of 58%. Seedlings were then thinned to one per pot at 1 week after emergence, and the plants were left to establish before they were shifted to a shadehouse in late autumn. The plants were kept in the shadehouse under natural light throughout winter. The average maximum and minimum temperatures during winter in the unheated glasshouse were 10.5˚C to 6.4˚C respectively. A susceptible population (OS) was collected from a site close to the Helensville Golf Club (36˚38'18.6"S 174˚30'23.2"E), but previous applications of any herbicides including clopyralid at this site were unlikely. The plants from population OS were also kept in the same shadehouse as population OR. Sine the resistant and susceptible plants flowered at different times, the cross-pollination between them was unlikely. The seeds from each population were collected at maturity in early summer and stored at 5˚C until the beginning of this research.

## Response to clopyralid

The response of population OR to clopyralid (Versatill, 300 g ae L$^{-1}$ as amine salt) was compared to population OS using a dose-response experiment. Plants of each population were established from seeds. For this, 30 seeds were placed in each planter bag (PB 5, 120mm x 120mm x 200mm, 3 L) filled with potting mix and fertilizer as described above. The pots were kept in a heated glasshouse under natural light at average daily max/min temperatures of 23.4/19.8˚C, and an average RH of 52%. At one week after emergence, the seedlings were thinned to 15 seedlings in each pot, and the plants were left to establish before they were sprayed at the 4–5 leaf stage with clopyralid. Clopyralid was applied at 0, 37.5, 75, 150, 300 (recommended rate), 600, 1200, 2400 and 4800 g ae ha$^{-1}$ in an initial dose-response experiment. The doses were chosen to cover the whole range of responses from no effect to complete death of plants [19]. The plants were treated with clopyralid using a laboratory track sprayer which delivered 230 L ha$^{-1}$ of spray solution at 200 kPa. The treated plants were then returned to the same glasshouse and kept for 4 weeks before evaluating the response of plants to herbicide treatments. The average daily maximum and minimum temperatures during the 4 weeks following treatment were 23.1 and 20.1˚C respectively, and the average RH was 59%. To evaluate the response of plants to clopyralid, the number of plants that survived the application was recorded to calculate the percentage of surviving plants for each rate. This experiment used a randomized design with four replicates (i.e. four pots) and then was repeated using the same method outlined above, but in the second dose-response experiment, plants were treated with 0, 37.5, 75, 150, 300, 600, 1200, 2400 and 4800, 9600 and 19200 g ae ha$^{-1}$ of clopyralid. Higher doses were added in the second experiment because the highest rate of clopyralid used in the first dose-response caused no mortality in the resistant phenotype.

## Cross-resistance to other synthetic auxin herbicides

The pattern of cross-resistance to MCPA (MCPA 750, 750 g ae L$^{-1}$ as the dimethylamine salt), picloram (Spike, 200 g ae L$^{-1}$ as amine salt), dicamba (Kamba 500, 500 g ae L$^{-1}$ as dimethylamine salt) and mecoprop-p (Duplosan KV, 600 g ae L$^{-1}$ as potassium salt of the optically active isomer) was evaluated for population OR and the response was compared to population OS

**Table 1. Herbicides that were applied to clopyralid-resistant (OR) and clopyralid susceptible (OS) populations in dose-response experiments.**

| Herbicide | Rate (g ae ha$^{-1}$) |
|-----------|------------------------|
| Picloram | 0, 25, 50, 100, 200, 400, 800, 1600 and 3200 |
| MCPA | 0, 93.75, 187.5, 375, 750, 1500, 3000, 6000 and12000 |
| Dicamba | 0, 100, 200, 400, 800, 1600, 3200 and 6400 |
| Mecoprop-p | 0, 150, 300, 600, 1200, 2400, 4800 and 9600 |

using the same dose-response experiment method as outlined above. Plants were established and grown as described above. The rates used for each herbicide are summarized in Table 1. This experiment used a randomized design with four replicates (i.e. four pots) and was conducted twice.

## Growth characteristics

Preliminary experiments reported that certain growth characteristics of clopyralid-resistant *S. sessilis* were different to susceptible plants [6]. Here, we quantified and compared the growth characteristics of the clopyralid-resistant with clopyralid-susceptible phenotypes of *S. sessilis*. Plants of each population were established from seeds using the method outlined above. At emergence, the seedlings were transplanted into pots (PB3, 100mm x 100mm x 200mm, 1.7 L) containing the same potting mix and slow-release fertilizer as those for the dose-response experiments with each pot containing only one seedling. The pots were kept under the same conditions as outlined for the dose-response experiments. At 40 days after emergence, the distance between the tips of the longest leaves either side of the rosette (rosette width) was measured in one direction to estimate the diameter of the plant rosette. A second measurement was then made perpendicular to the first measurement, and the two measurements were averaged for each plant. The plants were photographed before removing the plants (shoot plus root) from each pot. The photographs were used to study the morphological differences in leaflet lobes between resistant and susceptible phenotypes. The harvested plants were divided into root and shoot, and the root was washed with tap water. The leaf area of each harvested plant was measured using a digital leaf area meter (LiCor model-3100; LiCor, Lincoln, USA). The harvested shoot and root materials were oven-dried separately at 80°C for 48 h then weighed. This experiment consisted of eight replicates and was conducted twice.

## Evaluation of internal transcribed spacer regions

Internal transcribed spacer (ITS) regions are DNA markers that can be used to identify plant species [20]. The ITS of the clopyralid-resistant phenotype was evaluated and compared to those published for *S. sessilis* previously [21]. For this, initially, the genomic DNA was extracted from the leaves taken from clopyralid-resistant *S. sessilis* using a method described previously [22]. To amplify the ITS regions (ITS1 and ITS2), previously published primers [23] with some modifications were used. The forward (ITS-18SF: 5`-GAACCTTATCGTTTA GAGGAAGGAG-3`) and reverse (ITS-26R: 51`-AAGCCGCCCGATTTTCAAGC-3`) primers cover 840 bp portion of the *S. sessilis* ribosomal RNA gene (KX064030.1) flanking both ITS1 and ITS2 regions. Polymerase chain reaction (PCR) was used to amplify the ITS regions. The PCR reaction contained 12.5 µl of Q5Ⓡ high-fidelity 2X master mix (NEB, UK), 20 ng of DNA template, 0.4 µM of each forward and reverse primer and nuclease-free water to bring the volume of reaction to 25 µl. The PCR thermocycling program included initial denaturation at 98°C (one cycle of 30 s), denaturation at 98°C (35 cycles of 10 s), 35 cycles of 30 s annealing

at 55˚C, 35 cycles of 15 s extension at 72˚C, followed by one cycle final extension at 72˚C (2 min). The PCR products were then loaded on a 1x LB (lithium borate) 1% agarose gel (0.5 μg ml$^{-1}$ ethidium bromide) before they were run at 5 V cm$^{-1}$ for 0.5 h and visualized under UV illumination using a Gel Doc XR 2000 system (Bio-Rad Laboratories). The PCR products were then sequenced by the Massey Genome Service using the same forward and reverse primers outlined above and the DNA sequenced data were analyzed, assembled and compared using an online sequence alignment tool, Emboss Needle (https://www.ebi.ac.uk) [24]. The ITS region sequence from this study was compared to the ITS region sequences of *S. sessilis* (AM774471.1), *S. anthemifolia* (AY947414.1), *S. mutisii* (HE860705.1), and *S. stolonifera* (AJ864601.1) available in the data set using the basic local search alignment tool (Blast) (https://blast.ncbi.nlm.nih.gov/Blast.cgi), and the sequences with highest blast scores were considered best hit sequences. As ITS2 regions contain enough variability to distinguish closely related species [20], the secondary structures of ITS2 region was predicted and assessed on the ITS2 database web server (http://its2.bioapps.biozentrum.uni-wuerzburg.de) [25]. In addition, the Kimura-2-parameter (K2P) model was used to calculate genetic distance within interspecies using MEGA X software [26].

## Statistical analyses

The survival data from dose-response experiments were fitted to a three-parameter log-logistic model (Eq 1) after they were checked for normality (Shapiro–Wilk test) and homogeneity of variance (Levene's tests).

$$Y = \frac{d}{1 + \exp(b(\log x) - \log(LD_{50}))} \tag{1}$$

where Y is plant survival, d is the upper limit, x is the herbicide rate, $LD_{50}$ is the herbicide rate corresponding to 50% reduction in plant survival, and b is the slope around $LD_{50}$. The dose-response data were analyzed using the *drc* package in R v. 3.1.2. [27], and the $LD_{50}$ estimates of resistant and susceptible populations for each herbicide were compared using the '*compParm*' statement in the *drc* package [27]. The data from both dose-response experiments were analyzed separately due to the variability in response to herbicides between two runs. The data from the growth characteristic experiments were pooled as there was no significant difference between the two runs (p > 0.05). The differences in growth characteristics between populations OR and OS were statistically analyzed and compared using a Student's t-test at a 5% probability.

## Results

### Clopyralid dose-response experiments

The results from the first clopyralid dose-response experiment revealed a high level of resistance to clopyralid for OR population compared to the susceptible population (OS). While all OS plants treated at 300 g ae clopyralid ha$^{-1}$ were completely dead at 4 weeks after application, even 4800 g ae clopyralid ha$^{-1}$ did not cause any mortality in the plants of OR population (Fig 1A, Table 2). In the second clopyralid dose-response experiment, the range of clopyralid rates was further extended to generate a better dose-response curve for estimating the level of clopyralid resistance in population OR. All the plants of OS population were completely controlled at 150 g ae clopyralid ha$^{-1}$; however, there was only 18% mortality recorded for the plants of population OR treated at 19200 g ae clopyralid ha$^{-1}$ (Fig 1B, Table 2). Based on these results, it appeared that population OR was highly resistant to clopyralid with a level of resistance over 225-fold.

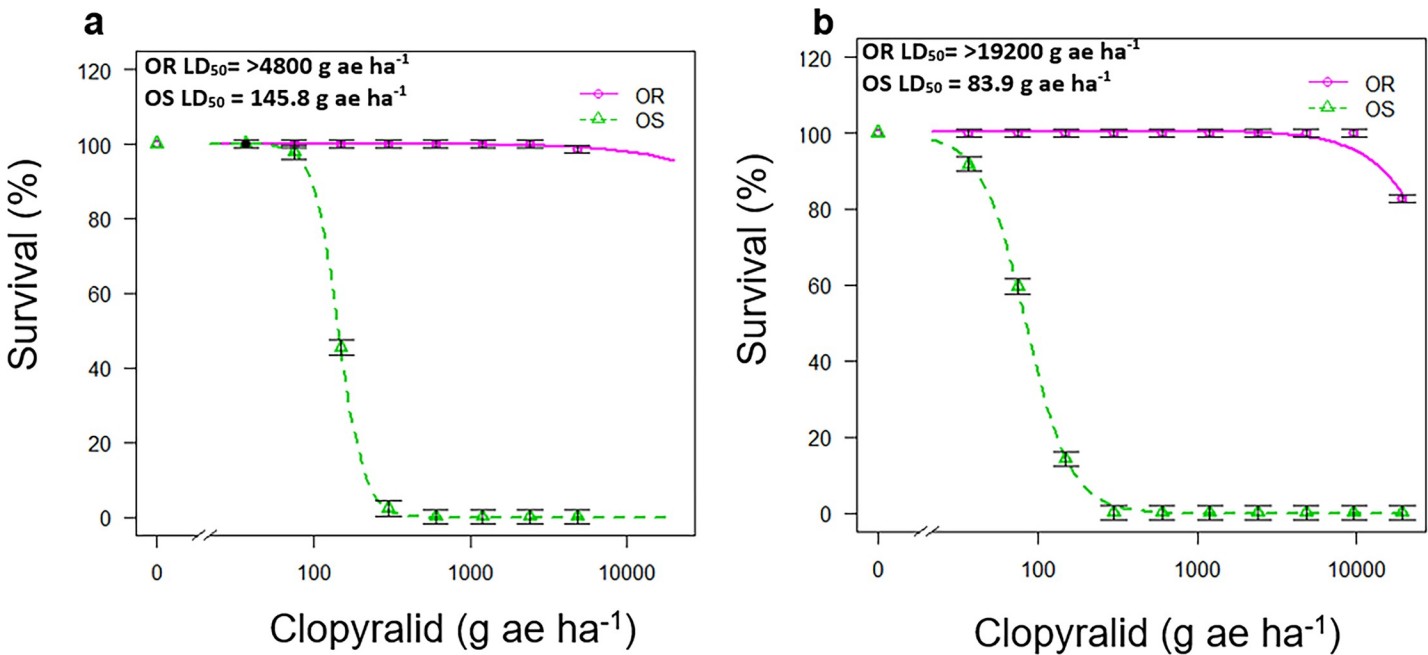

**Fig 1.** Fitted clopyralid dose-response curves for two *S. sessilis* populations, the resistant population OR and the susceptible population OS in (a) the first and (b) second dose-response experiments. The percentage of survival of treated plants was used to produce the fitted curves. Vertical bars represent ± standard error of the mean.

## Cross-resistance dose-response experiments

Results from dose-response experiments for other synthetic auxin herbicides showed that population OR was cross-resistant to picloram, MCPA and dicamba, but with different levels of resistance to each herbicide (Fig 2, Table 3).

All plants of population OS treated at 100 g ae picloram ha⁻¹ were completely controlled in both dose-response experiments, whereas 100% mortality was only recorded for population OR at 800 and 400 g ae picloram in the first and second dose-response experiments, respectively. Based on the values for 50% reduction in survival of individuals ($LD_{50}$ values), population OR was estimated to be 2.6- and 2.4-fold more resistant to picloram relative to population OS (Fig 2A and 2B).

**Table 2. Parameters (see footnote) estimated from the four-parameter log-logistic model analysis of clopyralid dose-response experiments for clopyralid-resistant (OR) and susceptible (OS) populations evaluated at 4 weeks after treatment.**

| First dose-response experiment | | | | |
|---|---|---|---|---|
| Population | d (±SE) | b (±SE) | $LD_{50}$ (±SE) | $LD_{50}$ RF |
| OR | 100 (0.6) | 1.4 (N/A) | >4800 (N/A) | >33.1 |
| OS | 100 (1.0) | 5.4 (0.8) | 145.0 (2.0) | |
| *P-value* | | | NA | |
| Second dose-response experiment | | | | |
| Population | d (±SE) | b (±SE) | LD50 (±SE) | R/S LD50 |
| OR | 100.5 (1.1) | 1.6 (0.4) | >19200 (N/A) | >225.9 |
| OS | 99.0 (2.9) | 3.2 (0.4) | 83.9 (3.8) | |
| *P-value* | | | N/A | |

d = the upper limit, b = the slope around the $LD_{50}$, $LD_{50}$ = the rate of herbicide (g ae ha⁻¹) required to cause 50% mortality, SE = standard error $LD_{50}$ RF = resistant/susceptible factor based on $LD_{50}$ ratios. N/A = not applicable.

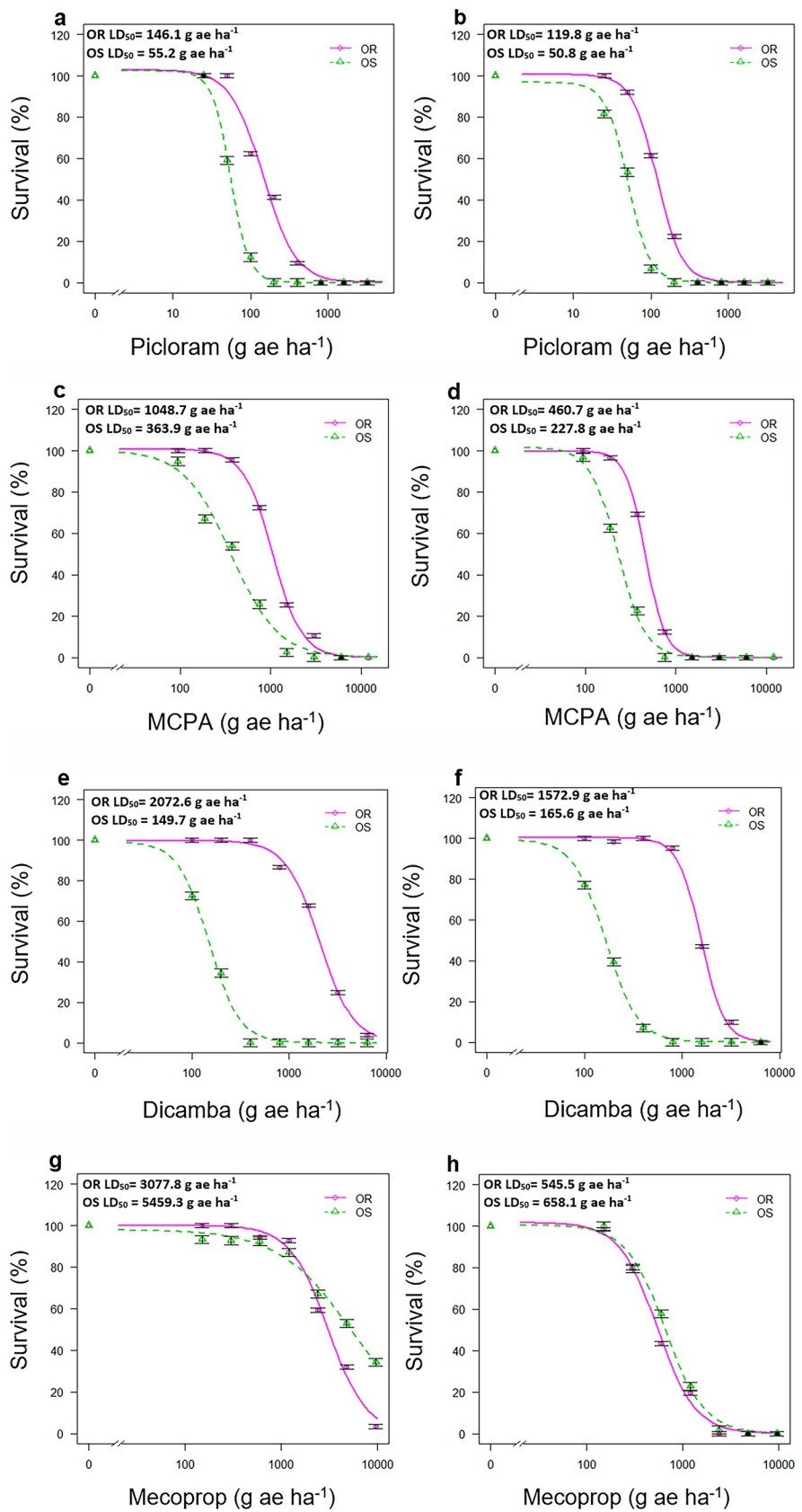

**Fig 2.** Fitted dose-response curves for two *S. sessilis* populations, the resistant population OR and the susceptible population OS in (a) first and (b) second picloram dose-response experiments, (c) first and (d) second MCPA dose-response experiments, e) first and (f) second dicamba dose-response experiments, and (g) first and (h) second mecoprop dose-response experiments. The percentage of survival of treated plants was used to produce the fitted curves. Vertical bars represent ± standard error of the mean.

Population OR also showed a low level of resistance to MCPA (Table 3). Based on the $LD_{50}$ values, population OR was found to be 2.9 and 2.1 times less sensitive to MCPA compared to population OS, in the first and second dose-response experiments, respectively (Table 3). MCPA application rates of 750 and 375 ae ha$^{-1}$ resulted in 100% mortality in all plants of population OS in the first and second dose-response experiments respectively, while greater rates of MCPA were needed to provide complete control of the individuals of population OR in both experiments (Fig 2C and 2D).

Population OR displayed a high level of resistance to dicamba in both dose-response experiments (Table 3). In both dose-response experiments, the individuals of population OS were

**Table 3. Parameters (see footnote) estimated from the four-parameter log-logistic model analysis of picloram, MCPA, dicamba and mecoprop dose-response experiments for clopyralid-resistant (OR) and susceptible (OS) populations evaluated at 4 weeks after treatment.**

| First dose-response experiment | | | | | |
|---|---|---|---|---|---|
| | Population | d (±SE) | b (±SE) | LD$_{50}$ (±SE) | LD$_{50}$ RF |
| Picloram | OR | 102.9 (2.6) | 2.1 (0.2) | 146.1 (10.1)a | 2.6 |
| | OS | 102.3 (2.9) | 3.7 (0.5) | 55.2 (2.4)b | |
| | *P-value* | | | *0.006* | |
| MCPA | OR | 100.7 (2.6) | 2.7 (0.4) | 1048.7 (64.8)a | 2.9 |
| | OS | 100.1 (4.5) | 1.7 (0.2) | 363.9 (37.1)b | |
| | *P-value* | | | *0.0008* | |
| Dicamba | OR | 99.8 (2.6) | 2.5 (0.5) | 2072.6 (232.0)a | 13.8 |
| | OS | 99.2 (5.0) | 2.7 (0.4) | 149.7 (12.9)b | |
| | *P-value* | | | *0.0001* | |
| Mecoprop-p | OR | 99.9 (2.6) | 2.2 (0.3) | 3077.8 (238.0)b | 0.6 |
| | OS | 98.0 (3.4) | 1.1 (0.2) | 5459.3 (708.6)a | |
| | *P-value* | | | *0.0001* | |
| Second dose-response experiment | | | | | |
| | Population | d (±SE) | b (±SE) | LD$_{50}$ (±SE) | LD$_{50}$ RF |
| Picloram | OR | 101.5 (2.0) | 2.7 (0.4) | 119.8 (4.3)a | 2.4 |
| | OS | 97.0 (2.8) | 3.0 (0.4) | 50.8 (2.1)b | |
| | *P-value* | | | *0.0001* | |
| MCPA | OR | 99.8 (1.8) | 3.9 (0.5) | 460.7 (15.3a) | 2.1 |
| | OS | 101.6 (2.5) | 2.8 (0.3) | 227.8 (9.7)b | |
| | *P-value* | | | *0.002* | |
| Dicamba | OR | 100.2 (1.2) | 3.7 (0.4) | 1572.9 (42.5)a | 9.5 |
| | OS | 99.4 (2.5) | 2.6 (0.2) | 165.6 (6.9)b | |
| | P-value | | | 0.0001 | |
| Mecoprop-P | OR | 101.7 (3.4) | 2.2 (0.2) | 545.5 (38.1)b | 0.8 |
| | OS | 100.5 (3.4) | 2.1 (0.3) | 658.1 (49.3)a | |
| | *P-value* | | | *0.04* | |

d = the upper limit, b = the slope around the $LD_{50}$, $LD_{50}$ = the rate of herbicide (g ae ha$^{-1}$) required to cause 50% mortality, SE = standard error $LD_{50}$ RF = resistant/susceptible factor based on $LD_{50}$ ratios. Different letters within one herbicide treatment indicate significant differences between the two populations, according to t-tests (P < 0.05).

completely dead at 800 g ae dicamba ha$^{-1}$, while at this dicamba rate, only 10% mortality was recorded for population OR (Fig 2E and 2F). The LD$_{50}$ values for population OR when treated with dicamba were found to be significantly greater than those of population OS in both dose-response experiments (Table 3). Population OR was 13.7-fold more resistant to dicamba than population OS, based on the LD$_{50}$ R/S ratio in the first run, and a 9.5-fold difference was recorded in the second run (Table 3).

Population OR was found to be more sensitive than population OS when treated with mecoprop-P (Fig 2G and 2H). Comparison of the LD$_{50}$ values showed significant differences between the populations, indicating a small negative cross-resistance to mecoprop in population OR (Table 3).

## Determination of differences in growth traits

To quantify the differences in growth characteristics between clopyralid-resistant and clopyralid-susceptible individuals, several traits were evaluated. At the cotyledon stage, there were no noticeable differences between the two populations (Fig 3A). However, differences in growth traits between the individuals of the two populations were evident with the appearance of true leaves. The leaflet shape differed between the two populations, with individuals of population OS having more lobes on each leaf than OR plants (Fig 3B and 3C).

When plants were compared at 40 days after emergence, the individuals of population OR were significantly larger as determined by their rosette width, leaf area and shoot dry weight (Table 4). There were also differences between both populations in their root size as the individuals of OR population had a larger root dry weight after 40 days of growth. Therefore, the total dry weight of population OR plants was also greater after 40 days than the OS plants. However, there were no significant differences in shoot/root ratios between the populations (Table 4).

## Comparison of sequence variation in ITS regions

The ITS primers used in this research successfully amplified the ITS regions. The results from the ITS1 region sequence alignment showed that the ITS1 region of population OR had 100% sequence homology to that of *S. sessilis* while it only had 94.5% sequence homology to the ITS1 region of *Soliva anthemifolia* and *Soliva mutisii* (Fig 4). Similarly, the ITS2 region sequence of population OR showed the greatest level of homology (99.60%) with the ITS2 region sequence of *S. sessilis* (Fig 5). The ITS2 region sequence of population OR had 95.50, 94.0 and 90.80% homology with that of *S. anthemifolia*, *S. mutisii* and *Soliva stolonifera*, respectively (Fig 5). Overall, these results showed that individuals of population OR had identical ITS region sequences with *S. sessilis* and the ITS2 region appeared to provide higher identification efficiency compared to the ITS1 region.

Differences in the ITS2 region sequence properties are shown in Table 5. The results showed that the individuals of population OR had the same guanine-cytosine (GC) content as *S. sessilis* and the interspecific genetic distance between *S. sessilis* and population OR was found to be very small ($1 \times 10^{-10}$), indicating that genetic information in ITS2 region sequences between these two are very close. The GC content in the other *Soliva* species was found to be lower than that of the individuals of population OR, with *S. stolonifera* having the lowest GC content. In addition, compared to *S. sessilis*, greater values were recorded for interspecific distance in the ITS2 region sequences of *S. anthemifolia*, *S. mutisii* and *S. stolonifera*, indicating that there was a great interspecies genetic distance between all three species and individuals of population OR. The ITS2 region secondary structures of these four *Soliva* species and the individuals of population OR are illustrated in Fig 6. Individuals of population OR

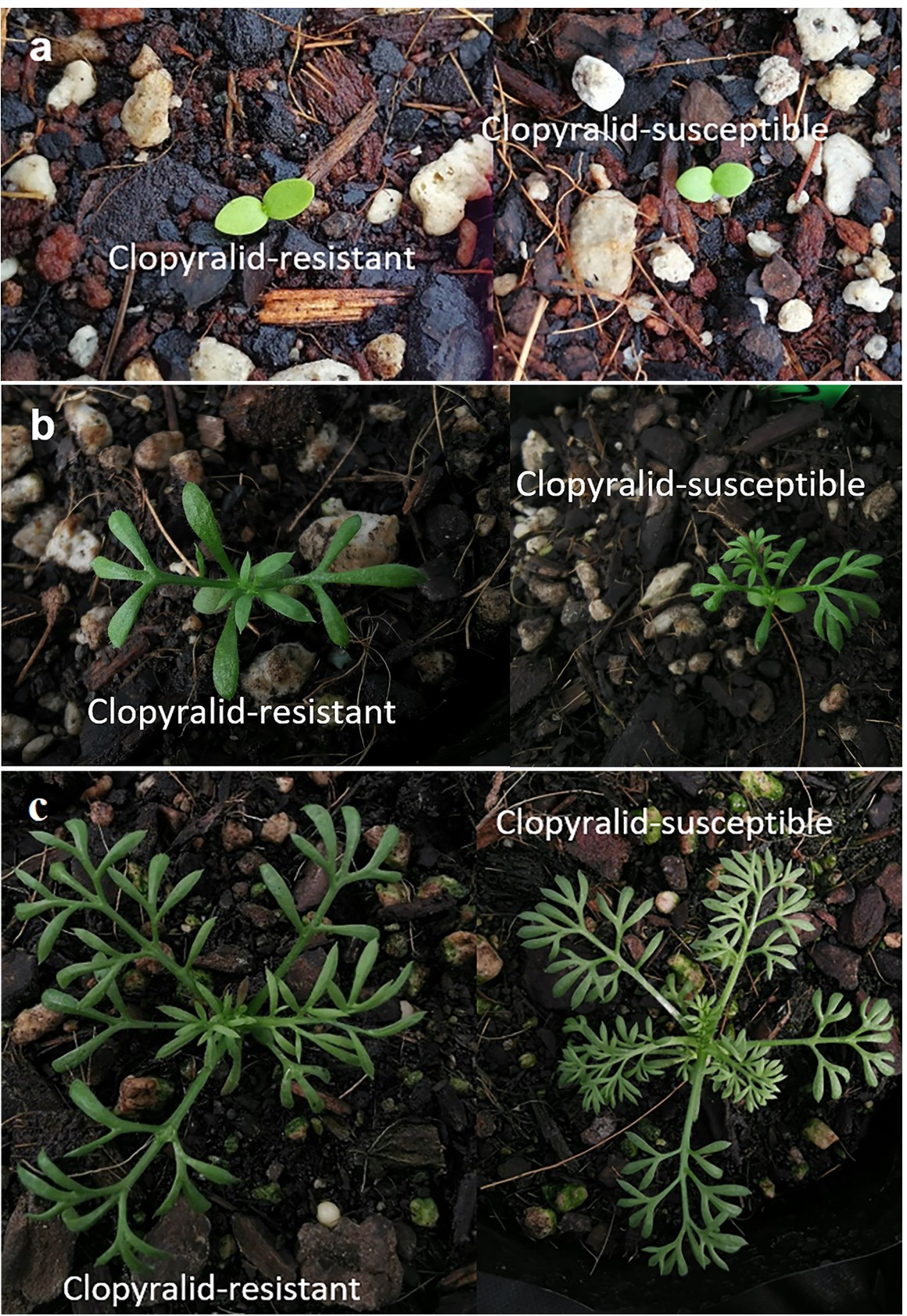

**Fig 3.** Variation in leaf morphology at (a) seedling (b) the 2–3 leaf and (c) the 5–6 leaf stage between clopyralid-resistant and clopyralid-susceptible plants.

had the same ITS2 region structure as that of *S. sessilis* (Fig 6A and 6B). However, there were several structural differences between population OR and other *Soliva* species in the ITS2 region secondary structure. For instance, there was a large bulge in the Helix II of individuals of population OR while *S. anthemifolia*, *S. mutisii* and *S. stolonifera* had a smaller "bulge" in the same region of Helix II (Fig 6A, and 6C–6E). In addition, there were differences in the number and position of the loops on Helices I and III between the ITS2 secondary structure of population OR and those of *S. anthemifolia*, *S. mutisii* and *S. stolonifera*. Taken together, these results revealed that based on the genetic information in ITS2 region, the individuals of population OR were conspecific with *S. sessilis*.

## Discussion

Globally, there are over 40 cases of resistance to synthetic auxin herbicide in weed species (both monocotyledon and dicotyledon) [4]. In New Zealand, there are five cases of resistance to synthetic auxin herbicides [28–30], of which *S. sessilis* is the only synthetic auxin herbicide-resistant species reported in turfgrass [5]. Clopyralid-resistant *S. sessilis* was initially reported in the early 2000s [5]; however, the level of resistance to clopyralid and the pattern of cross-resistance to other synthetic auxin herbicides were unknown in this resistant population. In this research, we recorded a very high level of resistance to clopyralid in the resistant phenotype. To the best of our knowledge, such a high level of resistance (> 225-fold) has not been reported for any of the weed species resistant to synthetic auxin herbicides. For instance, resistant phenotypes of *Bassia scoparia* and *Chenopodium album* were found to be 4.6-fold and 19-fold more resistant to dicamba respectively compared to their susceptible counterparts [28, 31]. Determining the level of resistance can hint at the potential mechanism of resistance [32]. The level of resistance to herbicides is also a function of other factors such as the type of mutation, the zygosity status of individuals for a specific mutation, and the number of mechanisms associated with resistance to a herbicide, with individuals that have accumulated multiple mechanisms of resistance displaying greater levels of herbicide resistance [33–36]. Taken together, the high level of clopyralid resistance in *S. sessilis* recorded in this research may suggest a different mechanism of resistance compared to that of other cases of synthetic auxin herbicide resistance, although the presence of multiple mechanisms of resistance cannot be ruled out.

Evaluating the pattern of cross resistance can inform us of alternative chemical options for managing herbicide resistance [37]. The pattern of cross-resistance can vary based on the herbicidal modes of action [37], and the type of the mutations associated with the mechanism of resistance [35, 38, 39]. The results of this research showed that clopyralid-resistant *S. sessilis* was cross-resistant to dicamba, picloram and MCPA but not mecoprop. In addition, the clopyralid-resistant phenotype had a high level of resistance only to dicamba, while the level of

**Table 4. Growth analysis of clopyralid-resistant (OR) and clopyralid -susceptible (OS) plants at 40 days after emergence.**

| Population | RosetteWidth (cm) | Leaf area (cm$^{-2}$) | Shoot dry weight (g) | Root dry weight (g) | Total dry weight (g) | Shoot/root ratio |
|---|---|---|---|---|---|---|
| OR | 11.2a | 24.2a | 0.144a | 0.0283a | 0.172a | 5.3 |
| OS | 8.5b | 14.7b | 0.103b | 0.0209b | 0.124b | 4.9 |
| *P-value* | *0.0001* | *0.0001* | *0.0001* | *0.005* | *0.0001* | *0.322* |

Mean values followed by different letters are significantly different between the two populations, according to t-tests (P < 0.05).

```
Population OR    TCGAACCCTGCAAAGCAGAACGACCCGCGAACATGTAACAACAACTTGTGTTGGGTGTAT
S. sessilis      TCGAACCCTGCAAAGCAGAACGACCCGCGAACATGTAACAACAACTTGTGTTGGGTGTAT
S. anthemifolia  TCGAACCCTGCAAAGCAGAACGACCCGTGAACATGTAACAACAACCTGTGTCGGGTGTAT
S. mutisii       TCGAACCCTGCAAAGCAGAACGACCCGTGAACATGTAACAACAACCTGTGTCGGGTGTAT
                 *************************** ****************** ***** *******

Population OR    CGAGCATTTGCTCGAGCCACCTGATGCTTTTGCCGATGTGCGTTCTTTGTGCCATTTTGG
S. sessilis      CGAGCATTTGCTCGAGCCACCTGATGCTTTTGCCGATGTGCGTTCTTTGTGCCATTTTGG
S. anthemifolia  CGAGCATTTGTTCGAGCCACCTGATGCTT-CGCCGATGTGCGTTCTTTGTGCCATTTTGG
S. mutisii       CGAGCATTTGTTCGAGCCACCTGATGCTT-TGCCGATGTGCGTTCTTTGTGCCATTTTGG
                 ********** ******************    ***************************

Population OR    CACTTGCGGACGTTGCAGCGGCACAATAACAAACCCCGGCACGGAACGTGCCAAGGAAAA
S. sessilis      CACTTGCGGACGTTGCAGCGGCACAATAACAAACCCCGGCACGGAACGTGCCAAGGAAAA
S. anthemifolia  CACTTGTGGATGTCGCAACGGCACGCTAACAAACCCCGGCACGGAACGTGCCAAGGAAAA
S. mutisii       CACTTGTGGATGTCGCAACGGCACACTAACAAACCCCGGCACGGAACGTGCCAAGGAAAA
                 ****** *** ** *** ******   *********************************

Population OR    CTAAACTTAAGAAGGGCTCTTTCCATGTGGCCTCGTTCGCGGTGTGCTCATGGGATGTGG
S. sessilis      CTAAACTTAAGAAGGGCTCTTTCCATGTGGCCTCGTTCGCGGTGTGCTCATGGGATGTGG
S. anthemifolia  CTAAACTTAAGAAGGGCTCTTTCCATGTTGCCTCGTTCGCGATGTGCTCATGGGATGTGG
S. mutisii       CTAAACTTAAGAAGGGCTCTTTCCATGTTGCCTCGTTCGCAATGTGCTCATGGGATGTGG
                 ****************************** **********   *****************

Population OR    CTTCTTTGTAATTACAAAC
S. sessilis      CTTCTTTGTAATTACAAA-
S. anthemifolia  CTGCTTTGTAATTAC----
S. mutisii       CTGCTTTGTAATTACAAA-
                 ** ***********
```

**Fig 4. Sequence alignment of the internal transcribed spacer 1 (ITS1) regions of clopyralid-resistant population (OR), *S. sessilis*, *S. anthemifolia*, and *S. mutisii*.** Hyphens (-) denote alignment gaps and asterisks donates residues conserved in all sequences.

resistance to MCPA and picloram was relatively low. Both clopyralid and picloram belong to the pyridine carboxylic acid class of synthetic auxin herbicides, while dicamba and MCPA belong to the benzoic acid and phenoxy acid classes, respectively.

The pattern and level of cross-resistance to different classes of synthetic auxin herbicides have been shown to vary for other resistant weed species [37]. For instance, picloram-resistant *Centaurea solstitialis* was found to be highly cross-resistant to clopyralid while it showed a low level of cross-resistance to dicamba and no cross-resistance to 2,4-D [40]. Dicamba-resistant *C. album* was highly cross-resistant to picloram and aminopyralid (pyridine carboxylic acids), but it was not cross-resistant to either 2,4-D or mecoprop (phenoxy acids) [41]. The varied patterns of cross-resistance to synthetic auxin herbicides in a dicamba-resistant phenotype of *B. scoparia* was associated with a single mutation within a highly conserved region of an AUX/indole-3-acetic acid (IAA) protein, IAA16 [42]. The dicamba-resistant phenotypes were cross-resistant to 2,4-D, picloram, fluroxypyr (pyridine carboxylic acid) and quinclorac (quinoline carboxylic acid) [42]. Varying patterns of cross-resistance recorded in this research and others can be attributed to different mechanisms or specific mutations associated with resistance to synthetic auxin herbicides. For example, a specific point mutation may only confer resistance to a class of herbicides or a small number of chemicals within a herbicide group [37].

```
Population OR    ATCGCGTCGCCCCCGCCTC--CATCCC-TGTCAAGGTATGTGATTCGGGGGCGGAGATTG
S. sessilis      ATCGCGTCGCCCCCGCCTC--CATCCC-TGYCAAGGTATGTGATTCGGGGGCGGAGATTG
S. anthemifolia  -TCGCGTCGCCCCCGACTCACCATCCCCTATCAAGGATTGTGATTCGGGGGCGGAGATTG
S. mutisii       ATCGCGTCGCCCCCTACTCACCATCCCCTATCAAGGATTGTGATTCGGGGGCGGAGATTG
S. stolonifera   ATCGTGTCGCCCCCGCATCACAATCAC-CGTTAAGGTGTGTGATTCGGGGAGCGGAGATTG
                 *** ********    **   *** *      ****  ********** **********

Population OR    GACTCCCGTGCTCATGGTGCGGTTGTCCAAAATAGGAGTCCCCTTCGATGGACGCACGAT
S. sessilis      GACTCCCGTGCTCATGGTGCGGTTGTCCAAAATAGGAGTCCCCTTCGATGGACGCACGAT
S. anthemifolia  GACTCCCGTGCCCATGGTGCGGTTGTCCAAAATAGGAGTCCCCTTCGATGGACGCACGAT
S. mutisii       GACTCCCGTGCCCATGGTGCGGTTGTCCAAAATAGGAGTCCCCTTYGATGGACGCACGAT
S. stolonifera   GACTCCCGTGCTTA-GCTGCGGTTGTCCAAAATAGGAGTCCCCTTCGATGGACGCACGAT
                 **********    *  * ***************************  **************

Population OR    TAGTGGTGGTCGTAAAAACCCTCGTCTTGTGTCGTGCGTTGTAAGTCGCGAGGGAACGAC
S. sessilis      TAGTGGTGGTCGTAAAAACCCTCGTCTTGTGTCGTGCGTTGTAAGTCGCGAGGGAACGAC
S. anthemifolia  TAGTGGTGGTCGTCAAAACCCTCGTCTTGTGTCGTGCGTTATAAGTCGCAAGGGAAGAAC
S. mutisii       TAGTGGTGGTCGTCAAAACCCTCGTSTNGTGTCGTGCGTTATAAGTCGCAAGGGAAGAAC
S. stolonifera   AAGTGGTGGTCGTCAAAACCCTCGTCTTGTGTCGTGTGTTATAAGTCGCAAGGGAAGAAC
                 ************ *********** *  * ********* ***  ******* ******  **

Population OR    TCTTCAAGAAACCCCAACGTGTTGTCTTATGACGACGCTTCGACCGCGACCCCAGGTCAG
S. sessilis      TCTTCAAGAAACCCCAACGTGTTGTCTTATGACGACGCTTCGACCGCGACCCCAGGTCAG
S. anthemifolia  TCTTCAAGAACCCCAAACGTGTTGTCTTGTGACAACGCTTCGACCGCGAC----------
S. mutisii       TCTTCAAGAACCCCAA-CGTGTTGTCTTGTGACGACGCTTCGAC----------------
S. stolonifera   TCTATACGAACCCCAA-TGTGTTGTCTTTAGACGATGCTTCGACC---------------
                 ***   * *** *** *   ********* *   *  ********

Population OR    GCGGGAC
S. sessilis      GCGGGAC
S. anthemifolia  -------
S. mutisii       -------
S. stolonifera   -------
```

**Fig 5. Sequence alignment of the internal transcribed spacer 2 (ITS2) regions of clopyralid-resistant population (OR), *S. sessilis*, *S. anthemifolia*, *S. mutisii* and *S. stolonifera*.** Hyphens (-) denote alignment gaps and asterisks donate residues conserved in all sequences.

**Table 5. Internal transcribed spacer 2 (ITS2) region properties of different Soliva species and clopyralid-resistant *S. sessilis* population (OR).**

| Species | Base number (bp) | GC%[#] | Genetic distance[*] |
|---|---|---|---|
| Population OR | 243 | 58.0 | - |
| *S. sessilis* | 243 | 58.0 | $10 \times e^{-10}$ |
| *S. anthemifolia* | 229 | 55.5 | 0.064 |
| *S. mutisii* | 220 | 54.5 | 0.067 |
| *S. stolonifera* | 222 | 51.8 | 0.13 |

[#] the percentage of guanine-cytosine content.

[*] Kimura-2-parameter (K2P) model was used to calculate genetic distance.

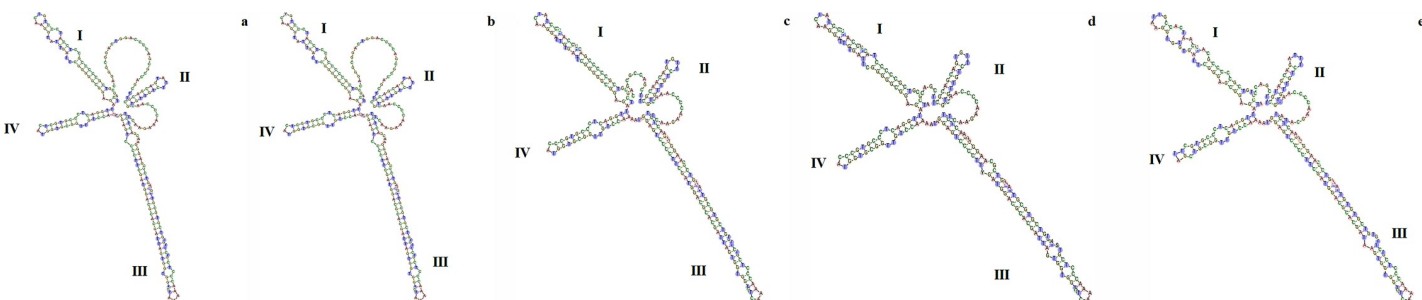

**Fig 6.** The predicted internal transcribed spacer 2 (ITS2) secondary structure of (a) clopyralid-resistant population (OR), (b) *S. sessilis*, (c), *S. anthemifolia*, (d) *S. mutisii* and (e) *S. stolonifera*. The four helices are labelled I–IV. The secondary structures were predicted and assessed using the ITS2 database web server (http://its2.bioapps. biozentrum.uni-wuerzburg.de).

Although these results suggest no cross resistance to mecoprop in the resistant *S.sessilis* plants, this herbicide has generally been considered to be poor at controlling *S. sessilis* [43]. Herbicides registered for use in New Zealand to control this weed in turfgrass that contain mecoprop are either mixtures with ioxynil and bromoxynil, or mixtures with MCPA and dicamba [15]. Data in Fig 2 shows that good control was only achieved at rates exceeding 1000 g ae ha$^{-1}$ of mecoprop, the optically active isomer, which is equivalent to 2000 g ae ha$^{-1}$ of the normal mecoprop present in many turfgrass herbicides used in New Zealand. The highest recommended rate of the mixture with MCPA and dicamba is needed to reach a rate of 2000 g ae ha$^{-1}$ of mecoprop, and the cross-resistance to dicamba means there will be little assistance from this component. Mecoprop is not used alone as a turfgrass herbicide as it does not control a particularly wide range of weed species [15]. Most other turfgrass herbicides in New Zealand make use of clopyralid, triclopyr and picloram to control *S. sessilis*, all of which are not suitable for the resistant phenotype. Earlier research showed that the resistant phenotype can be controlled by the mecoprop + ioxynil + bromoxynil formulation available in New Zealand and also bentazone [6]. Both herbicides need to be applied while the seedlings are young though to get good control [15], so are less versatile than herbicides such as clopyralid that would normally be used in spring on older plants. It will probably be necessary for turfgrass managers with this resistant winter annual weed to depend more on keeping the turfgrass competitive during autumn to avoid germination rather than relying just on herbicide applications in spring as currently occurs.

In this study, variations in growth traits between clopyralid-resistant and clopyralid-susceptible phenotypes of *S. sessilis* were recorded. The results show that the clopyralid-resistant plants had fewer lobes on their leaves, but they were larger compared to their susceptible counterparts. Such distinct growth characteristics can be used by turfgrass managers for identifying the clopyralid-resistant *S. sessilis*. Variations in growth traits have been observed for *Arabidopsis* lines with mutations within their auxin receptor proteins [44]. Phenotypic variations have also been recorded for other synthetic auxin herbicide- resistant weeds. For instance, dicamba-resistant *C. album* phenotypes were found to be shorter, more branched and their leaves were less jagged compared to the wild-type [45]. Picloram-resistant phenotypes of *Sinapis arvensis* were found to have a serrated leaf margin while their susceptible counterparts had a smooth leaf margins [46]. The dicamba-resistant phenotype of *B. scoparia* was found to be shorter and had more ovate leaf blades compared to the susceptible ones [42]. The growth characteristics observed in the dicamba-resistant phenotype of *B. scoparia* were attributed to a single mutation within *IAA16* gene [42]. The mechanism associated with resistance to dicamba and picloram in *C. album* [47] and *S. arvensis* [48] phenotypes outlined above has not been

completely elucidated but non-target site mechanisms (herbicide enhanced metabolism and reduced herbicide absorption/translocation) were not associated with the resistance to dicamba [37] and picloram [48] in either species. Therefore, it is likely that a mutation in an auxin receptor protein [17] is associated with the mechanisms of resistance and the growth traits manifested by each of these resistant phenotypes. For instance, it is known that mutations in the Auxin/indole-3-acetic acid (Aux/IAA) transcription factors, *IAA9*, *axr5-1/IAA1*, *shy2/IAA3*, *axr2/IAA7*, *IAA16* and *IAA28* result in abnormalities in leaf shape and development [49].

ITS regions have been used as barcodes in plant taxonomy to identify plant species [50]. In order to confirm if the individuals of population OR were correctly identified as *S. sessilis*, we amplified the ITS region sequence of the individuals of population OR and the resultant sequence was compared to those of other *Soliva* species available in the National Center for Biotechnology Information (NCBI) database. The results showed that individuals of population OR shared the same ITS sequence as that of *S. sessilis*. Also, the results from interspecies genetic distances and the ITS2 secondary structure provided further evidence that the individuals of population OR are conspecific with *S. sessilis*.

## Conclusion

In conclusion, the results from this research confirm a very high level of resistance to clopyralid in *S. sessilis*. An assessment of the extent and level of cross-resistance to other synthetic auxin herbicides recommended for weed management in turfgrass showed that only mecoprop had no cross-resistance to this clopyralid-resistant phenotype. The greatly reduced number of lobes on each leaf associated with clopyralid-resistance can be used by turfgrass managers to detect the resistant plants and manage them accordingly. Future studies will involve evaluating the extent of the problem in turfgrass areas in New Zealand, understanding the mode of inheritance and investigating the molecular basis of resistance to clopyralid in *S. sessilis*.

## Author Contributions

**Conceptualization:** Hossein Ghanizadeh, Kerry C. Harrington.

**Data curation:** Hossein Ghanizadeh.

**Formal analysis:** Hossein Ghanizadeh.

**Funding acquisition:** Hossein Ghanizadeh, Kerry C. Harrington.

**Investigation:** Hossein Ghanizadeh, Fengshuo Li.

**Methodology:** Hossein Ghanizadeh, Kerry C. Harrington.

**Software:** Lulu He.

**Validation:** Hossein Ghanizadeh, Lulu He.

**Visualization:** Fengshuo Li.

**Writing – original draft:** Hossein Ghanizadeh.

**Writing – review & editing:** Kerry C. Harrington.

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
