## [Decision Letter · Decision Letter 0]

11 May 2021

PONE-D-21-11484

Characterization of clopyralid resistance in Soliva sessilis

PLOS ONE

Dear Dr. Ghanizadeh,

Thank you for submitting your manuscript to PLOS ONE. After careful consideration, we feel that it has merit but does not fully meet PLOS ONE’s publication criteria as it currently stands. Therefore, we invite you to submit a revised version of the manuscript that addresses the points raised during the review process.

The manuscript by Ghanizadeh et al. reports results from original studies characterizing the resistance against herbicide clopyralid in an important turfgrass weed, Soliva sessilis. The reviewers have provided constructive comments that are generally in favour of the manuscript. However, there are certain limitations highlighted by all the reviewers that I strongly encourage authors to address comprehensively. In particular, authors are encouraged to address some of the critical comments made by Reviewer 1. Should authors disagree with any of reviewers' comments they must provide strong justification for their arguments in a response letter. In addition to reviewers' comments, I also have following points that I request authors to consider while revising their manuscript:

- Better you also provide the most commonly used common name of the weed species in title as well.

- Add brief results on morphological differences in abstract; also add a sentence on management implications of your study in abstract

- Use either the term 'turfgrass' or 'turf' for consistency

- Provide GPS coordinates of sites where populations were collected

- Why call susceptible and resistant populations OS and OR? Why not just use susceptible and resistant

- Justify the use of different dose ranges between two runs of clopyralid response exp

- Not sure if missed - provide details of pot size/vol in each study

- What is Student`s t-test? Is it something different to commonly used LSD or Tuckey's test? If yes, any specific reason for using this?

- I would prefer adding LD50 values within figures. Also consider using colors or different marker shapes for figures as they not quite distinct or maybe photo quality in pdf is not that good?

- Strengthen the discussion by providing more reasoning for your results instead of comparing it with other studies.

We look forward to receiving your revised manuscript.

Kind regards,

Ali Bajwa, Ph.D

Academic Editor

PLOS ONE

Journal Requirements:

2. Please include a copy of Table 1 which you refer to in your text on page 7.

3. We note you have included two tables which you refer in the text of your manuscript, however both are labelled as Table 2. Please ensure that you label each Table by a separate number in the title and also cite the relevant table number in your text; if accepted, production will need this reference to link the reader to each Table.

Reviewers' comments:

Reviewer's Responses to Questions

**Comments to the Author**

1. Is the manuscript technically sound, and do the data support the conclusions?

Reviewer #1: Yes

Reviewer #2: Yes

Reviewer #3: Yes

2. Has the statistical analysis been performed appropriately and rigorously? 

Reviewer #1: Yes

Reviewer #2: Yes

Reviewer #3: Yes

3. Have the authors made all data underlying the findings in their manuscript fully available?

Reviewer #1: Yes

Reviewer #2: Yes

Reviewer #3: Yes

4. Is the manuscript presented in an intelligible fashion and written in standard English?

Reviewer #1: Yes

Reviewer #2: Yes

Reviewer #3: Yes

5. Review Comments to the Author

Reviewer #1: The current paper describes a new population of Soliva sessilis resistant to several auxinic herbicides. Resistant plants grew more than susceptible for most of the parameters analyzed, root/shoot ration was an exception, and an ITS analysis was used to confirm that both resistant and susceptible populations belonged to the same species.

In the present work, the dose responses were properly performed, and the results show a high level of resistance. On the growth assays, it would be important to measure the number of seeds produced in plants of both populations, since reproductive fitness cost is an important parameter for weed resistance establishment. Another experiment comparing differences of number of leaf lobes between populations would be necessary to support some the statements presented in the discussion. The ITS analysis can be treated more as supplementary data (specially Figures 4 and 5) because it is just confirming the correct species classification between both S. sessilis populations. I do not know how difficult is to perform crosses in this species, but it would be important to add a third section in this paper to characterize the mode of inheritance for the resistance trait.

The paper format and written content need to be improved. The language used in some sections is quite informal for a scientific journal and the content is a bit repetitive along the different sections. The introduction and can be more concise.

In the abstract the words “resistant” and “resistance” are used too many times. Additionally, there is a wrong use of the concept of evolution in the abstract - Herbicides do not cause "evolution", indeed, their continuous application can select resistant individuals that naturally occur in weed populations.

The identification of new cases of herbicide resistance in weeds is important for the weed science discipline, since it increases our knowledge on the process of microevolution on how intense application of herbicides along the years selects and pressures the establishment of weed resistant populations and how those genetic changes interfere on weed control, competition and reproduction. Studying those populations may help to create alternative practices of weed control that may be more effective in suppressing the infestation resistant populations in the place they were found and to spread to new areas. The identification of new populations opens new opportunities to study the molecular mechanisms of herbicide resistance that were not observed or did not occur in model species that herbicide targets were first identified. In this paper, it was identified auxinic herbicide resistant populations of S. sessilis however, as I already mentioned, more information related to weed development and mode of resistant inheritance would be necessary to attend the quality requirements required by the journal it was submitted.

Reviewer #2: The study was done to characterize clopyralid resistance in a Soliva sessilis population from New Zealand. The research methods used in the study including dose-response, comparing growth traits with susceptible population, and comparing sequence variation with other weeds is appropriate. Overall, good work by the researchers and a well-written paper. However, some minor issues need to be addressed and some findings need to be better discussed especially focusing on the implication of such findings. Comments are listed below:

Abstract: Please add one or two sentences on the implication of the findings of the study.

Line 25-27: Repetitive sentence, does not add any value to the abstract.

Line 107-108: Were the OR and OS plants covered by pollination bags to make sure there is no cross-pollination?

Line 125: Please mention that this was used to calculate percent survival.

Line 136: Table 1 is absent in the manuscript. Please add it.

Line 189: Three-parameter log-logistic regression model: Instead of writing it as R50, I suggest writing ‘e’ and then describe what ‘e’ is? or explain R50 as LD50.

Line 190-191: x is herbicide dose, e or R50 is the effective dose of herbicide needed to reduce the plant survival by 50% i.e., LD50

L194: Please mention that because of variability the dose-response runs for each herbicide were analyzed separately.

Line 219: 400 g ae ha-1 of picloram or 400 g ae picloram ha-1

Line 262: Please add full scientific name when mentioning the species for the first time (same for other species)

Line 303-305: What are the implications?

Line 312-319: Clopyralid and picloram belong to the same sub-group of auxins similarly MCPA and mecoprop both belong to phenoxycayboxylic acid group...It's interesting to see that within herbicides of the same sub-group there is such difference in response. e.g., OR is >225 fold resistant to clopyralid but has much lower level resistance to picloram. Any comment on such difference within the same sub-group?

L349-351: What are the implications of OR accumulating more biomass than OS?

L524: Two tables are labeled as ‘Table 2’. Please label as 2 and 3 (also in the text)

Table 2: Mecoprop dose-response 1: Interestingly, the obtained LD50 of both OR and OS are very high. I am guessing it's way higher than the field commended rate. Any comment on that?

Table 3: Instead of ‘total dry weight’ write ‘Total dry weight’

Reviewer #3: Please find my comments below:

L16: “highly resistant…”

L17-20: I suggest indicating the rates of each herbicide used. Were there numerous rates or a single rate of each herbicide evaluated?

L27-29: Mention some morphological trait differences between the R and S plants.

L29: What are the implications or significance of the findings? Why is it important to know that this weed species is cross-resistant to different herbicides?

L118: Any justification on how the rates were selected for this and other herbicides?

L146-148: It is not clear what “diameter” means. Are you referring to the area covered by the plant? Also, can you explain how the measurements were taken 90 degrees to each other, 90 degrees with respect to what?

L147-153: The traits measured do not necessarily seem to be classified as morphological traits rather than growth characteristics. I suggest referring to these traits as growth characteristics. To me, morphological traits would be leaf shape, leaf angle, stem diameter, height, flower color, pubescence, etc.

L248: I now see morphological traits (leaflet shape) that were not mentioned in the methodology.

L291-302: This seems to belong in the Introduction.

L374: There is a lack of discussion on the significance of the findings and what, if any, recommendations are available now that this weed species is found to be cross-resistant.

6. PLOS authors have the option to publish the peer review history of their article (what does this mean?). If published, this will include your full peer review and any attached files.

Reviewer #1: No

Reviewer #2: No

Reviewer #3: **Yes: **Te Ming Tseng

---

## [Author Response · Author response to Decision Letter 0]

27 May 2021

All the modifications have been indicated using the track changes feature in MS Word. Please note that the line number in the revised version is different from the original version submitted in the beginning due to the changes made in the text to address the comments. We therefore used the new line number where appropriate. Detailed responses to the reviewers are:

Editor

- Better you also provide the most commonly used common name of the weed species in title as well.

Response: A common name was included in the title. The common name was chosen according to Dr Ian Heap`s website of Global Herbicide-Resistant Weeds. 

- Add brief results on morphological differences in abstract; also add a sentence on management implications of your study in abstract

Response: A sentence was added to the abstract stating morphological differences between resistant and susceptible plants (L 27-28). The management implication of our research has been included in the abstract (L 38-42)

- Use either the term 'turfgrass' or 'turf' for consistency

Response: We used the term “turfgrass” in the entire manuscript.

- Provide GPS coordinates of sites where populations were collected

Response: GPS coordinates have been provided (L 108 and 120)

- Why call susceptible and resistant populations OS and OR? Why not just use susceptible and resistant

Response: Currently, we are looking for other cases of clopyralid resistance in other populations of Soliva sessilis. If we call the populations investigated in this research “resistant” and “susceptible” then it can be confusing later for the readers when we want to refer to these populations and compare them with new clopyralid-resistant populations, since all the populations will be resistant. However, by giving a specific label to each population, we can avoid such confusion. Therefore, we decided not to change the name of the populations. We hope that Editor agrees with our opinion.

- Justify the use of different dose ranges between two runs of clopyralid response exp

Response: A sentence was added to justify different dose ranges used between two runs (L 145-147). 

- Not sure if missed - provide details of pot size/vol in each study

Response: This information has been included (L 135 and L 173).

- What is Student`s t-test? Is it something different to commonly used LSD or Tuckey's test? If yes, any specific reason for using this? 

Response: The Student`s t-test (or the t-test) is a statistical analysis for comparing two sets of data. The LSD or Tukey`s test, however, are used when one wants to compare the mean of three or more sets of data. In our case, there are only two sets (resistant versus susceptible) of data. So the t-test is the appropriate analysis approach. 

- I would prefer adding LD50 values within figures. Also consider using colors or different marker shapes for figures as they not quite distinct or maybe photo quality in pdf is not that good?

Response: The graphs are in color and the LD50 values have been added to each figure. 

- Strengthen the discussion by providing more reasoning for your results instead of comparing it with other studies.

Response: The Discussion has been improved by discussing our results and providing further justification in different places (L 389-390, 394-397, 418-419, 436-441, 462-464). 

Editorial requirements

 Response: The manuscript has been prepared according to the requirements of the journal.

2. Please include a copy of Table 1 which you refer to in your text on page 7.

 Response: The Table 1 has been included and referred.

3. We note you have included two tables which you refer in the text of your manuscript, however both are labelled as Table 2. Please ensure that you label each Table by a separate number in the title and also cite the relevant table number in your text; if accepted, production will need this reference to link the reader to each Table.

Response: There is now only one table labelled as Table 2.

Reviewers

Reviewer 1

The current paper describes a new population of Soliva sessilis resistant to several auxinic herbicides. Resistant plants grew more than susceptible for most of the parameters analyzed, root/shoot ration was an exception, and an ITS analysis was used to confirm that both resistant and susceptible populations belonged to the same species.

In the present work, the dose responses were properly performed, and the results show a high level of resistance. 

Response: We would like to thank this peer-reviewer for the comments. 

On the growth assays, it would be important to measure the number of seeds produced in plants of both populations, since reproductive fitness cost is an important parameter for weed resistance establishment.

Response: The primary objective of growth analysis experiments was to obtain some quantitative data on the morphological differences observed between resistant and susceptible plants. We reckoned that by including these quantitative data the readers would have a better understanding of the differences. We did not mention anything, anywhere in the manuscript, about fitness differences between the populations. However, we suggested that these morphological differences can be used as markers for detecting the resistant plants. That is why we did not evaluate the number of seeds produced by each phenotype.

 Another experiment comparing differences of number of leaf lobes between populations would be necessary to support some the statements presented in the discussion. 

Response: Initially, we submitted some photos along with this manuscript as supplementary files showing what we meant by differences in leaf lobes between both phenotypes. However, we assumed that this reviewer overlooked the supplementary file. We have decided to include those photos as a Figure (Fig 3), so it will not be missed by the readers. We are not aware of any experimental techniques for assessing the leaf lobes and we noted that in all the manuscripts published to date, the authors used photos to show the differences in leaf lobes in the studied plants for example,

Blein, T., Pautot, V., & Laufs, P. (2013). Combinations of Mutations Sufficient to Alter Arabidopsis Leaf Dissection. Plants (Basel, Switzerland), 2(2), 230–247. https://doi.org/10.3390/plants2020230

The ITS analysis can be treated more as supplementary data (specially Figures 4 and 5) because it is just confirming the correct species classification between both S. sessilis populations. 

Response: We disagree with this comment. These figures should not be supplementary because the information presented in both figures are of considerable importance, and most people may miss it, as they rarely take the trouble to look at supplementary materials. Please note that this reviewer missed an important supplementary figure submitted with the original version in the beginning. 

I do not know how difficult is to perform crosses in this species, but it would be important to add a third section in this paper to characterize the mode of inheritance for the resistance trait.

Response: We agree that the information about the mode of inheritance is of importance but unfortunately, for species with a high level of self-pollination and minute flowers such as S. sessilis, it is very difficult to investigate the mode of inheritance (that is why the mode of inheritance in most cases, has been investigated for cross-pollination herbicide-resistant plants with large flowers). Our attempts so far of any pair-crossing between resistant and susceptible plants have failed. However, we have not given up yet. Having said that, even if we manage to pair-cross the phenotypes, it will at least take 1.5 years to investigate the mode of inheritance as it involves creation of F1 and BC/F2 generation for which we need to grow plants, cross them, collect seeds, grow plants out of those collected seeds, and then spray them to evaluate their responses. We have suggested that this mode of inheritance for clopyralid resistance will be investigated (L 481).

The paper format and written content need to be improved. The language used in some sections is quite informal for a scientific journal and the content is a bit repetitive along the different sections. The introduction and can be more concise.

Response: The language of the manuscript, format and the Introduction section have been improved.

In the abstract the words “resistant” and “resistance” are used too many times. 

Response: Although it might sound that these two terms have been repeated in different places in the manuscript, but they are no alternative terms that can be used interchangeably to these terms. 

Additionally, there is a wrong use of the concept of evolution in the abstract - Herbicides do not cause "evolution", indeed, their continuous application can select resistant individuals that naturally occur in weed populations.

Response: We had not used the term “cause” in the beginning; however, we have reworded the sentence for clarification (L 15-16). 

The identification of new cases of herbicide resistance in weeds is important for the weed science discipline, since it increases our knowledge on the process of microevolution on how intense application of herbicides along the years selects and pressures the establishment of weed resistant populations and how those genetic changes interfere on weed control, competition and reproduction. Studying those populations may help to create alternative practices of weed control that may be more effective in suppressing the infestation resistant populations in the place they were found and to spread to new areas. The identification of new populations opens new opportunities to study the molecular mechanisms of herbicide resistance that were not observed or did not occur in model species that herbicide targets were first identified. In this paper, it was identified auxinic herbicide resistant populations of S. sessilis however, as I already mentioned, more information related to weed development and mode of resistant inheritance would be necessary to attend the quality requirements required by the journal it was submitted.

Response: Thanks.

Reviewer 2

Reviewer #2: The study was done to characterize clopyralid resistance in a Soliva sessilis population from New Zealand. The research methods used in the study including dose-response, comparing growth traits with susceptible population, and comparing sequence variation with other weeds is appropriate. Overall, good work by the researchers and a well-written paper. However, some minor issues need to be addressed and some findings need to be better discussed especially focusing on the implication of such findings. Comments are listed below:

Response: We would like to thank this reviewer for the constructive comments that helped improve the manuscript.

Abstract: Please add one or two sentences on the implication of the findings of the study.

Response: This information has been added (L 38-42).

Line 25-27: Repetitive sentence, does not add any value to the abstract.

Response: The sentence has been deleted.

Line 107-108: Were the OR and OS plants covered by pollination bags to make sure there is no cross-pollination?

Response: The resistant and susceptible plants flowered at different times so, covering the plants was unnecessary (L 127-128). Also, according to our observation, this species is primarily a self-pollinating one.

Line 125: Please mention that this was used to calculate percent survival.

Response: It has been added (L148)

Line 136: Table 1 is absent in the manuscript. Please add it.

Response: The Table 1 has been added (L164).

Line 189: Three-parameter log-logistic regression model: Instead of writing it as R50, I suggest writing ‘e’ and then describe what ‘e’ is? or explain R50 as LD50.

Response: The formula has been improved (L223).

Line 190-191: x is herbicide dose, e or R50 is the effective dose of herbicide needed to reduce the plant survival by 50% i.e., LD50

Response: This has been modified (L224).

L194: Please mention that because of variability the dose-response runs for each herbicide were analyzed separately.

Response: This has been mentioned (L228-229).

Line 219: 400 g ae ha-1 of picloram or 400 g ae picloram ha-1

Response: “400 g ae ha-1 of picloram” is correct.

Line 262: Please add full scientific name when mentioning the species for the first time (same for other species)

Response: The full scientific names have been added (L329).

Line 303-305: What are the implications?

Response: Sentences have been added (L 389-390 and 394-397).

Line 312-319: Clopyralid and picloram belong to the same sub-group of auxins similarly MCPA and mecoprop both belong to phenoxycayboxylic acid group...It's interesting to see that within herbicides of the same sub-group there is such difference in response. e.g., OR is >225 fold resistant to clopyralid but has much lower level resistance to picloram. Any comment on such difference within the same sub-group?

Response: different patterns of cross-resistance are often recorded even for the herbicides that belong to the same sub-class, and it mostly has something to do with the type of the point mutation. We have added a sentence to address this (L 419-421).

L349-351: What are the implications of OR accumulating more biomass than OS?

Response: Accumulation of more biomass implies that the OR plants are bigger in size. As outlined above, our main purpose to evaluate the biomass was to provide quantitative data for the readers to get a better understanding of how big the resistant plants are. This information can be used as morphological markers for detecting the resistant plants as stated in L445-446.

L524: Two tables are labeled as ‘Table 2’. Please label as 2 and 3 (also in the text)

Response: This has been modified.

Table 2: Mecoprop dose-response 1: Interestingly, the obtained LD50 of both OR and OS are very high. I am guessing it's way higher than the field commended rate. Any comment on that?

Response: As stated in L 431, his herbicide usually does not give full control of S. sessilis when it is applied alone. However, out of curiosity, we wanted to know if the mechanism conferring clopyralid resistance could also confer resistance to mecoprop. However, since no cross-resistance to mecoprop was observed in the clopyralid-resistant S. sessilis, this herbicide can still be used in mixture with other non-auxinic herbicides as stated in L-435.

Table 3: Instead of ‘total dry weight’ write ‘Total dry weight’

Response: This has been modified.

 Reviewer 3

Reviewer #3: Please find my comments below:

Response: We would like to thank this reviewer for his constructive comments.

L16: “highly resistant…”

Response: This has been addressed (L18).

L17-20: I suggest indicating the rates of each herbicide used. Were there numerous rates or a single rate of each herbicide evaluated?

Response: According to PLOS ONE editorial requirements, the abstract must not exceed 300 words. As we had to address various comments from reviewers and Editor within the 300-word limit, we decided to prioritise the comments and address the ones that are more important. We appreciate that it would be informative if we could include the rates, but we think the reads can always refer to the Materials and Methods section for such information. Since this journal is an open-access one, there will be not limited access for anybody at all times to the Materials and Methods section. 

L27-29: Mention some morphological trait differences between the R and S plants.

Response: This has been added (L 27-28).

L29: What are the implications or significance of the findings? Why is it important to know that this weed species is cross-resistant to different herbicides?

Response: This has been discussed in L 38-42.

L118: Any justification on how the rates were selected for this and other herbicides?

Response: A sentence has been added to address this (L 141-142).

L146-148: It is not clear what “diameter” means. Are you referring to the area covered by the plant? Also, can you explain how the measurements were taken 90 degrees to each other, 90 degrees with respect to what?

Response: The term “diameter” has been replaced with “rosette width” (L 177). Also, a sentence has been added to state how the measurements were taken (L 176-178).

L147-153: The traits measured do not necessarily seem to be classified as morphological traits rather than growth characteristics. I suggest referring to these traits as growth characteristics. To me, morphological traits would be leaf shape, leaf angle, stem diameter, height, flower color, pubescence, etc.

Response: The terms “morphological traits” and “ morphological characteristics” have been replaced with “growth traits” and “growth characteristics”, respectively, where appropriate as suggested by this reviewer.

L248: I now see morphological traits (leaflet shape) that were not mentioned in the methodology.

Response: We photographed the plants to show the differences in leaf shape. This has been addressed in L 180-183.

L291-302: This seems to belong in the Introduction.

Response: This part has been shifted to the Introduction section (L 86-91).

L374: There is a lack of discussion on the significance of the findings and what, if any, recommendations are available now that this weed species is found to be cross-resistant.

Response: The implications and the significance of the results reported in this manuscript have been addressed in different places in the manuscript as requested by the Editor and other reviewers. The chemical options for managing this resistant species in turfgrass have been addressed and discussed in L422-432 and 434-436.

---

## [Decision Letter · Decision Letter 1]

16 Jun 2021

Characterization of clopyralid resistance in lawn burweed (Soliva sessilis)

PONE-D-21-11484R1

Dear Dr. Ghanizadeh,

We’re pleased to inform you that your manuscript has been judged scientifically suitable for publication and will be formally accepted for publication once it meets all outstanding technical requirements.

Kind regards,

Ali Bajwa, Ph.D

Academic Editor

PLOS ONE

Additional Editor Comments (optional):

The reviewers and I have now assessed the revised manuscript and we are satisfied with the revision undertaken by authors. Therefore, I am pleased to recommend the acceptance of this manuscript; however, I request authors to incorporate a couple of minor changes suggested by Reviewer 1 at proofs stage. Those suggested changes can be seen below as well as in Reviewer-1's comments.

The manuscript has greatly improved since the first version. Here are some suggestions:

Line 27-28: “…and more reliance on avoiding germination using turfgrass competition may be needed.”. Probably this phrase needs an introductory idea, something like:

“… considering the new challenges, other integrated management practices may be adopted such as using turfgrass to reduce weed germination”.

Line 39: On the phrase “evolution of herbicide resistance in weeds was inevitable”; I would suggest to change it for “the occurrence of herbicide resistance in weed populations was inevitable”

Lines 43-68: The introduction is too long; I would suggest making that section more concise.

Line 224: “in order to” can be removed.

Line 435: It’s not just IAA9, but also axr5-1/IAA1, shy2/IAA3, axr2/IAA7, iaa16 and iaa28.

Reviewers' comments:

Reviewer's Responses to Questions

**Comments to the Author**

1. If the authors have adequately addressed your comments raised in a previous round of review and you feel that this manuscript is now acceptable for publication, you may indicate that here to bypass the “Comments to the Author” section, enter your conflict of interest statement in the “Confidential to Editor” section, and submit your "Accept" recommendation.

Reviewer #1: All comments have been addressed

Reviewer #2: All comments have been addressed

2. Is the manuscript technically sound, and do the data support the conclusions?

Reviewer #1: Yes

Reviewer #2: Yes

3. Has the statistical analysis been performed appropriately and rigorously? 

Reviewer #1: Yes

Reviewer #2: Yes

4. Have the authors made all data underlying the findings in their manuscript fully available?

Reviewer #1: Yes

Reviewer #2: Yes

5. Is the manuscript presented in an intelligible fashion and written in standard English?

Reviewer #1: Yes

Reviewer #2: Yes

6. Review Comments to the Author

Reviewer #1: The manuscript has greatly improved since the first version. Here are some suggestions:

Line 27-28: “…and more reliance on avoiding germination using turfgrass competition may be needed.”. Probably this phrase needs an introductory idea, something like:

“… considering the new challenges, other integrated management practices may be adopted such as using turfgrass to reduce weed germination”.

Line 39: On the phrase “evolution of herbicide resistance in weeds was inevitable”; I would suggest to change it for “the occurrence of herbicide resistance in weed populations was inevitable”

Lines 43-68: The introduction is too long; I would suggest making that section more concise.

Line 224: “in order to” can be removed.

Line 435: It’s not just IAA9, but also axr5-1/IAA1, shy2/IAA3, axr2/IAA7, iaa16 and iaa28.

Reviewer #2: The authors have addressed my comments and have improved the manuscript. Missing table has been added and table 2 has been renumbered.

7. PLOS authors have the option to publish the peer review history of their article (what does this mean?). If published, this will include your full peer review and any attached files.

Reviewer #1: No

Reviewer #2: No

---

## [Editor Report · Acceptance letter]

21 Jun 2021

PONE-D-21-11484R1 

Characterization of clopyralid resistance in lawn burweed (*Soliva sessilis*) 

Dear Dr. Ghanizadeh:

I'm pleased to inform you that your manuscript has been deemed suitable for publication in PLOS ONE. Congratulations! Your manuscript is now with our production department. 

Kind regards, 

on behalf of

Dr. Ali Bajwa 

Academic Editor

PLOS ONE